# Fezf2-positive fork cell-like neurons in the mouse insular cortex

**Manabu Taniguchi**[1], **Misaki Iwahashi**[1], **Yuichiro Oka**[1,2], **Sheena Y. X. Tiong**[1,2,3], **Makoto Sato**[1,2,4]*

**1** Department of Anatomy and Neuroscience, Graduate School of Medicine, Osaka University, Suita, Japan, **2** Molecular Brain Science, Division of Developmental Neuroscience, Department of Child Development, United Graduate School of Child Development (UGSCD), Osaka University, Suita, Japan, **3** Faculty of Science, Institute of Biological Sciences, University of Malaya, Kuala Lumpur, Malaysia, **4** Graduate School of Frontier Biosciences, Osaka University, Suita, Japan

* makosato@anat2.med.osaka-u.ac.jp

**Data Availability Statement:** All relevant data are within the paper and its Supporting Information files.

**Funding:** This work was supported by the Japan Society for the Promotion of Science (JSPS)

## Abstract

The fork cell and von Economo neuron, which are found in the insular cortex and/or the anterior cingulate cortex, are defined by their unique morphologies. Their shapes are not pyramidal; the fork cell has two primary apical dendrites and the von Economo neurons are spindle-shaped (bipolar). Presence of such neurons are reported only in the higher animals, especially in human and great ape, indicating that they are specific for most evolved species. Although it is likely that these neurons are involved in higher brain function, lack of results with experimental animals makes further investigation difficult. We here ask whether equivalent neurons exist in the mouse insular cortex. In human, Fezf2 has been reported to be highly expressed in these morphologically distinctive neurons and thus, we examined the detailed morphology of Fezf2-positive neurons in the mouse brain. Although von Economo-like neurons were not identified, Fezf2-positive fork cell-like neurons with two characteristic apical dendrites, were discovered. Examination with electron microscope indicated that these neurons did not embrace capillaries, rather they held another cell. We here term such neurons as holding neurons. We further observed several molecules, including *neuromedin B* (*NMB*) and *gastrin releasing peptide* (*GRP*) that are known to be localized in the fork cells and/or von Economo cells in human, were localized in the mouse insular cortex. Based on these observations, it is likely that an equivalent of the fork cell is present in the mouse.

## Introduction

The insular cortex (insula) is an integral brain hub that is reciprocally connected with the sensory, emotional, motivational and cognitive systems. It receives afferents from the dorsal thalamus and several sensory cortical areas, and projects to the frontal cortical areas and the subcortical brain regions, which include the nucleus accumbens and the caudate putamen that are implicated in motivation [1–5]. The insular cortex is segregated into three different areas: the granular, dysgranular and agranular insular cortices (abbreviated as GI, DI and AI, respectively), which differ in their cytoarchitectures. The GI has a classical six-layered structure, with

KAKENHI Grant Numbers 15K15015, 17H04014, and 2H03414 to MS. No. The funders had no role in study design, data collection and analysis, decision to publish, or preparation of the manuscript.

**Competing interests:** The authors have declared that no competing interests exist.

**Abbreviations:** AI, agranular insular cortex; bvFTD, behavioral variant of frontotemporal dementia; Cl, claustrum; DI, dysgranular insular cortex; ec, external capsule; Fezf2, forebrain embryonic zinc finger 2; GI, granular insular cortex; GRP, gastrin releasing peptide; ISHH, *in situ* hybridization histochemistry; NMB, neuromedin B; PBS, phosphate buffered saline; PFA, paraformaldehyde; RFP, red fluorescent protein; SSC, saline sodium citrate; TDP-43, TAR DNA-binding protein 43; VEN, von Economo neuron.

a well-developed granular layer IV. In the DI, layer IV is thinner and layer V is more prominent than in the GI. Unlike the GI and DI, no obvious layer IV can be recognized in the AI, whereas layers II, III, V and VI are well developed [6–8]. The three main areas of the insular cortex are strongly interconnected along the dorso-ventral and rostro-caudal axes [9].

A unique feature of the insular cortex in higher animals, such as human and great ape, is the presence of special cells in layer V, the von Economo neurons (VENs) and fork cells [10–13]. VENs had large spindle-shaped perikarya with a thick basal dendrite and an apical dendrite [13]. The fork cell was first reported as a distinct neuronal type in the human insular cortex [14] and it closely resembles the VEN, but has two apical dendrites instead of a single apical dendrite [11,12]. These types of neurons had been found in the anterior cingulate cortex and the insular cortex in mammalian phylogeny, especially in large-brained and social animals, including hominoid primates. Up to now, VENs have been identified in distantly related mammals such as cetaceans [15], perissodactyls [16] and elephants [17], as well as in humans, great apes and macaques [10,18,19]. However, it is generally assumed that such cells do not exist in rodents, the one most commonly used laboratory animal. If the analogous neurons exist in the experimental animal, it would present a unique platform to study these cell types in detail.

Some VENs and fork cells are found to be positive for TDP-43 aggregation at the earliest stages of sporadic behavioral variant of frontotemporal dementia (bvFTD), and VENs and fork cells are considered to be involved in bvFTD [20–22]. However, an immense lack of appropriate laboratory animals has notably impeded the progress of studies on VENs and fork cells. Neuronal circuits of the VENs and fork cells in human remain unknown and the biological identity of these neurons are unspecified.

In the developing human neocortex, there exist higher animal-specific neurons, such as outer radial glia. Even though it is assumed that such cells are not present in laboratory animals, recent studies revealed that mice have prototypes of the progenitors in the cortex [23,24]. Therefore, we set out to explore the possibility that equivalents of VENs and/or fork cells exist in the mouse cortex.

Although VENs and fork cells are defined based on their morphological features, data on the molecular characteristics of VENs and/or fork cells have been accumulating. In humans, there are several proteins that are selectively expressed in VENs [10,25–28]. Among these proteins, neuromedin B (NMB), gastrin releasing peptide (GRP) and Fezf2 showed the most intense staining in VENs.

We asked whether these specific molecules for VENs and/or Fork cells were expressed in the mouse insular cortex. Interestingly, we found that some neurons in layer V of the insular cortex were positive for *NMB*, *GRP* or Fezf2. Moreover, A few Fezf2-positive neurons exhibited morphology representing that of fork cell-like neuron. This is the first study reporting the presence of fork cell-like neurons in the mouse insular cortex.

## Materials and methods

### Animals

ICR (SLC, MGI Cat# 5462094, RRID:MGI:5462094) mice were used. Fezf2-tdTomato mouse (STOCK Tg(Fezf2-tdTomato) SZ89Gsat/Mmucd, ID: 036540-UCD) was obtained from the Mutant Mouse Regional Resource Center, a NCRR-NIH funded strain repository, and was donated to the MMRRC by the NINDS funded GENSAT BAC transgenic project. The day of birth was designated as postnatal day 0 (P0). All experiments were conducted in compliance with the guidelines for the use of laboratory animals of Osaka University, and approved by Animal Research Committee of Osaka University. All possible efforts were made to minimize the number of animals used and their suffering.

## Probe generation

cDNA fragments of *NMB* and *GRP* were PCR-amplified from mouse brain cDNA with the following primer pairs. *NMB*-fwd.: 5′ `AAGCAAGATTCGAGTGCACC` 3′; rev.: 5′ `CAGCATCCGG TTTGTTCCAT` 3′; *GRP*-fwd.: 5′ `TGAATCCCCGTCCCTGTATG` 3′; and rev.: 5′ `GGTAGCAAA TTGGAGCCCTG` 3′. Amplified fragments were cloned into pGEM-T vector (Promega). *In vitro* transcription of cRNA probes was performed with T7 or SP6 RNA polymerase (Roche) and RNA DIG labeling mix (Roche) according to the manufacturer's instructions using the template plasmids linearized with an appropriate restriction enzyme.

## Tissue preparation

During all surgical procedures, animals were deeply anesthetized by intraperitoneal injection of a combination anesthetic (MMB: 0.3 mg/kg of medetomidine, 4.0 mg/kg of midazolam, and 5.0 mg/kg of butorphanol) and intracardially perfused with ice-cold phosphate buffered saline (PBS) followed by 4% paraformaldehyde (PFA) in PBS. Whole brains were carefully dissected out after perfusion and post-fixed in 4% PFA overnight at 4°C, then transferred to 30% sucrose solution ($\geq$24 h or until brains sank to the bottom of the tube at 4°C). The whole brains were embedded in OCT compound. The tissue was then cryosectioned at 16 µm, mounted on SuperFrost microslide glasses, and stored at −80°C. For electron microscopy, animals were transcardially perfused with ice-cold PBS followed by 0.05% glutaraldehyde and 4% paraformaldehyde in PBS. Brains were removed and post-fixed in the same fixative for 2h at 4°C, followed by immersion in 0.1 M PB overnight at 4°C. Brain sections of 40 µm thick were then cut by vibrating microtome and stored at 4°C.

## Nissl staining

The sections were placed in 0.5% cresyl violet in water at RT for 5 min, then were briefly rinsed twice in 95% ethanol and dipped in 100% ethanol three times before being dehydrated in xylene three times for 5 min each. The sections were affixed to glass coverslips using Entellan new solution (Merck KGaA, Darmstadt, Germany) and examined under a light microscope.

## *in situ* Hybridization Histochemistry (ISHH)

*In situ* hybridization was performed as described before [29] using cryosections (16 µm) prepared from wild-type ICR at P56 as mentioned above. The cryosections were air dried for 1 h and fixed in 4% PFA in PBS for 10 min at room temperature. The sections were then incubated in 0.2 M HCl for 10 min, followed by permeabilization with Proteinase K (7.9 µg/ml; Roche) digestion for 10 min at 37°C. Next, the sections were treated with acetic anhydride in 0.1 M triethanolamine for 10 min. The slides were rinsed with PBS in between each step. Finally, the sections were transferred to 5× saline sodium citrate (SSC) for 10 min or longer. Hybridization was carried out with the generated probes in the hybridization buffer (50% formamide, 5× SSC, 200 µg/ml yeast tRNA) overnight for at least 16 h at 55°C. High-stringency washes were carried out in the following steps: 5× SSC, 20 min at room temperature; 2× SSC, 20 min at 65°C; two washes with 0.2× SSC, 20 min at 65°C and lastly, the slides were transferred to PBS at room temperature. Detection of specific hybridization was performed using anti-Digoxigenin coupled with alkaline phosphatase (Roche), and subsequently visualized using nitro blue tetrazolium chloride/5-bromo-4-chloro-3-indolyl-phosphate (NBT/BCIP). Sense probes were used as negative controls and no signals were observed with the sense probes.

## Immunohistochemistry (IHC)

Brain cryosections were prepared from Tg(Fezf2-tdTomato) mice (P56). After being air-dried for an hour at room temperature, the sections were treated in PBS. Sections were then blocked for 1 h with 5% BSA, 0.1% TritonX-100 in PBS, and then incubated overnight at 4°C with anti RFP (1:400; rabbit; MBL International, PM005, RRID: AB_591279). This was followed by incubation with donkey anti-rabbit IgG Alexa Fluor 568 (Thermo Fisher Scientific, Waltham, MA, USA) or biotinylated anti- rabbit IgG secondary antibody (Vector Laboratories Inc., Burlingame, CA) for 4–6 h at 4°C. All secondary antibodies were diluted at 1:500 in PBS. Fluorescence signals were imaged with a laser scanning confocal microscope (LSM5 PASCAL with Zeiss). For detection of biotinylated secondary antibody binding sites, sections were visualized using a Vectastain Elite ABC kit according to manufacturer's manual (Vector Laboratories Inc., Burlingame, CA). After amplification with avidin-biotin complex from the Vectastain Elite ABC kit, reaction products were visualized with 0.05 M Tris-HCl buffer (TBS; pH 7.6) containing 0.05% diaminobenzidine tetrahydrochloride (DAB) and 0.01% hydrogen peroxide.

## ISHH-IHC dual staining method

Shortly after the ISHH signal detection was completed, the sections were treated in PBS and then processed for Fezf2 IHC staining as described above.

## Immunoelectron microscopy

Brain sections were prepared from Tg(Fezf2-tdTomato) mice (P56). IHC was performed according to staining procedure with diaminobenzidine (DAB) [30]. In brief, Tg(Fezf2-td-Tomato) mice (P56) were transcardially perfused with ice-cold PBS followed by 0.05% glutaraldehyde and 4% paraformaldehyde in PBS. Brains were removed and post-fixed in the same fixative for 2h at 4°C, followed by immersion in 0.1 M PB overnight at 4°C. Brains sections of 40 μm thick were then cut by vibrating microtome. Immunohistochemistry was performed using free-floating sections according to the ABC method. The anti-RFP antibodies (1:400; rabbit; MBL International, PM005, RRID: AB_591279) were used at a dilution of 1:400 as a primary antibody. Biotinylated anti-rabbit IgG (Vectastain Elite) was used as a secondary antibody. Immunoreactivity was visualized with 0.05% diaminobenzidine and 0.01% hydrogen peroxide in 50 mM Tris, pH 7.6. Sections were washed several times in a 0.1 M phosphate buffer (pH 7.4) and immersed in 1% osmium tetroxide in 0.1 M PB for 1 hour on ice, followed by three rinses with distilled water. After the third rinse, the samples were stained with 0.5% uranyl acetate in distilled water overnight at 4°C. The samples were then dehydrated in increasing concentrations of ethanol (65%, 75%, 85%, 95% and 100%) at RT, with 5 min incubation time between ethanol changes. They were then dehydrated three times with anhydrous ethanol processed by molecular sieve and twice with propylene oxide at room temperature (RT), with 20 min incubation time. The samples were then replaced with a 1:1 solution of propylene oxide: epoxy resins overnight at RT. This mixture was thoroughly removed and they were flat-embedded in Epon 812. After polymerization, ultrathin sections were cut using a Reichert-Nissei Ultracut N microtome (Leica, Germany) and observed with a transmission electron microscope (H-7650, Hitachi, Tokyo, Japan) without counterstaining with uranyl acetate and lead citrate.

## 3D images

Confocal Z stack images were acquired using a Zeiss LSM 810 confocal microscope and Zen software (Carl Zeiss). These images were collected at 1024 × 1024 pixel resolution. For 3D

rendering, images were collected at 8 bit with the optimal slice settings. Compiled software of Zeiss microscope was used for image analysis and 3D rendering.

### The Allen Mouse Brain Atlas ISHH Data

Data from the Allen Mouse Brain Atlas (http://mouse.brain-map.org/) on male 56-day-old C57BL/6J mice were used for analyses in this study.

## Results

Whereas VENs and fork cells are defined based on their morphologies, molecules that are expressed in VENs and/or fork cells endow such neurons with specific characteristics. Table 1 is the summary of recent reports on molecules that are expressed in VENs and/or fork cells.

We asked whether VENs and fork cells or their equivalents exist in the mouse brain based on the molecules that are expressed, in addition to the morphologies. Based on these reports in Table 1, we first studied the expression profiles of Fezf2 in the mouse insular cortex. Since

**Table 1.**

| Reported human VEN and/or fork cell marker molecules |
| --- |
| VENs/fork cells |
| vesicular monoamine transporter 2 (VMAT2) * [26] |
| *vesicular monoamine transporter 2 (VMAT2)* * [26] |
| gamma-aminobutyric acid (GABA) receptor subunit (GABRQ) * [26] |
| *gamma-aminobutyric acid (GABA) receptor subunit (GABRQ)* * [26,31] |
| adrenoreceptor α-1A (ADRA1A) * [26] |
| *adrenoreceptor α-1A (ADRA1A)* * [26,31] |
| neuromedin B (NMB) * [10] |
| gastrin releasing peptide (GRP) * [10] |
| interleukin 4 receptor, alpha (IL4Rα) [27] |
| activating transcription factor 3 (ATF3) [27] |
| *bone morphogenetic protein 3 (BMP3)* * [31] |
| *POU class 3 homeobox 1 (POU3F1)* * [31] |
| VENs |
| vasopressin 1a receptor (V1aR) [25] |
| serotonin 2b receptor (5-HT2BR) [25] |
| dopamine d3 receptor (D3R) [25] |
| disrupted in schizophrenia 1 (Disc1) * [10] |
| *chicken ovalbumin upstream promoter transcription factor-interacting protein 2 (CTIP2)* * [32] |
| vesicle amine transport 1 like (VAT1L) [28] |
| *vesicle amine transport 1 like (VAT1L)* [28] |
| carbohydrate sulfotransferase 8 (CHST8) [28] |
| *carbohydrate sulfotransferase 8 (CHST8)* [28] |
| LY6/PLAUR domain containing 1 (LYPD1) [28] |
| *LY6/PLAUR domain containing 1 (LYPD1)* [28] |
| sulfatase 2 (SULF2) [28] |
| *sulfatase 2 (SULF2)* [28] |
| *forebrain embryonic zinc finger 2 (FEZF2)* * [31,32] |
| *integrin, alpha 4 (ITGA4)* * [31] |

* expression in the insular cortex is reported.
Italics indicate mRNA.

tdTomato expression is driven by Fezf2 promoter in Fezf2-tdTomato mouse, Fezf2-tdTomato mouse allowed us to visualize whole cell morphology including axons and dendrites comparable to Golgi staining. We used Fezf2-tdTomato mouse for the identification of von Economo neurons and/or folk cells.

## A few Fezf2-positive fork cell-like neurons were identified in the mouse insular cortex

Since a great majority of VENs are identified in layer V of the human insular cortex, we focused on this layer for further investigation. In order to study the detailed morphology of Fezf2-positive cells, we performed immunohistochemistry using antibody against RFP with frozen brain sections of Tg(Fezf2-tdTomato) mouse. In the mouse insular cortex, Fezf2-positive neurons were distributed widely in the rostral-caudal axis of layers V/VI, where most neurons were densely located in layer V (GI and DI) (Figs 1 and 2). Unexpectedly, a few

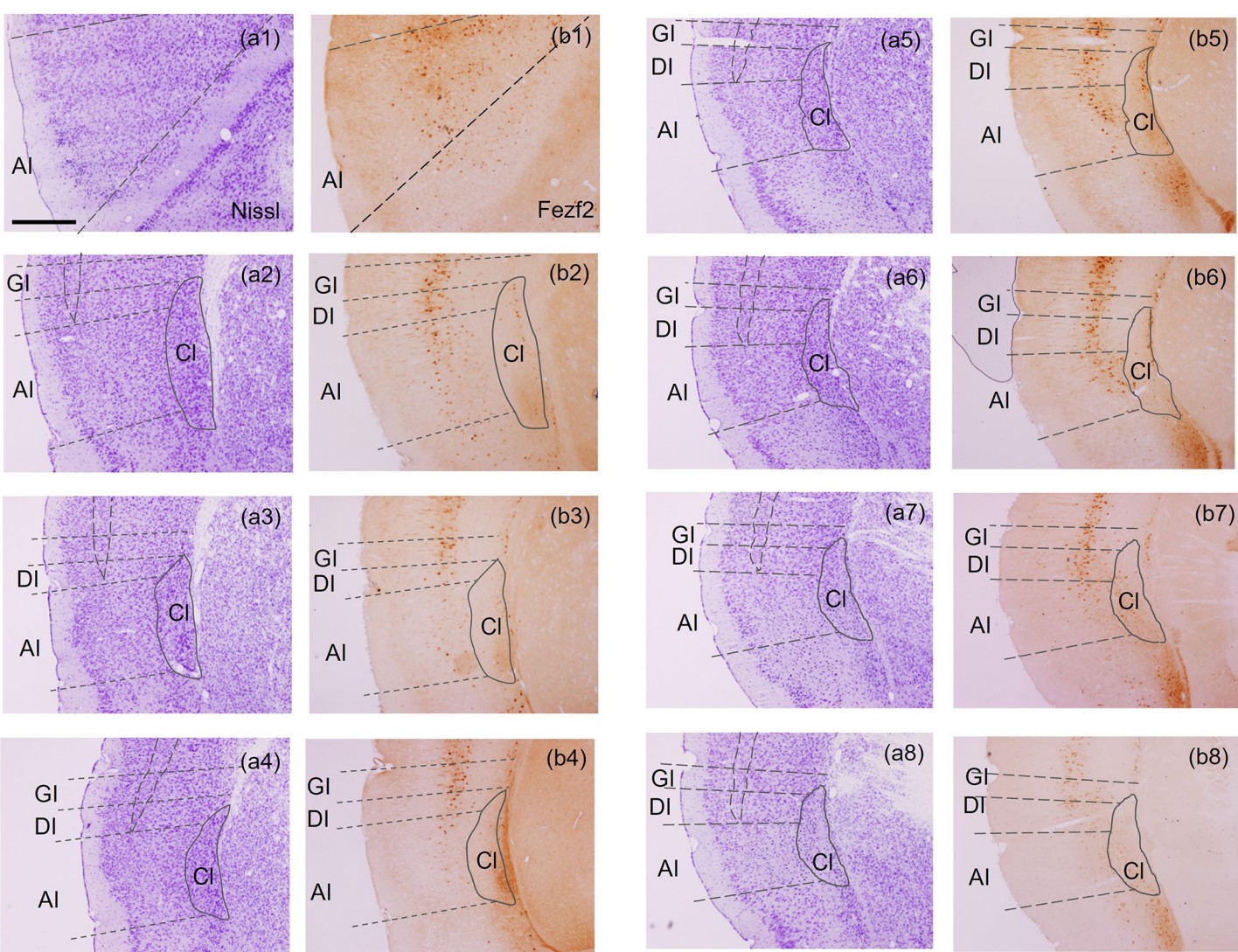

**Fig 1. Fezf2 was expressed in layer V/VI neurons of the mouse insular cortex.** Sections were numbered in rostral-caudal order. (a1)–(a8); Nissl staining. (b1)–(b8); adjacent tissue sections of (a1)–(a8). Fezf2 was expressed widely in rostral-caudal axis in layers V/VI. Fezf2 was visualized by immunohistochemistry against RFP in Tg (Fezf2-tdTomato) mice. Dotted lines indicate the borders of GI, DI and AI. Scale bar: 200 µm.

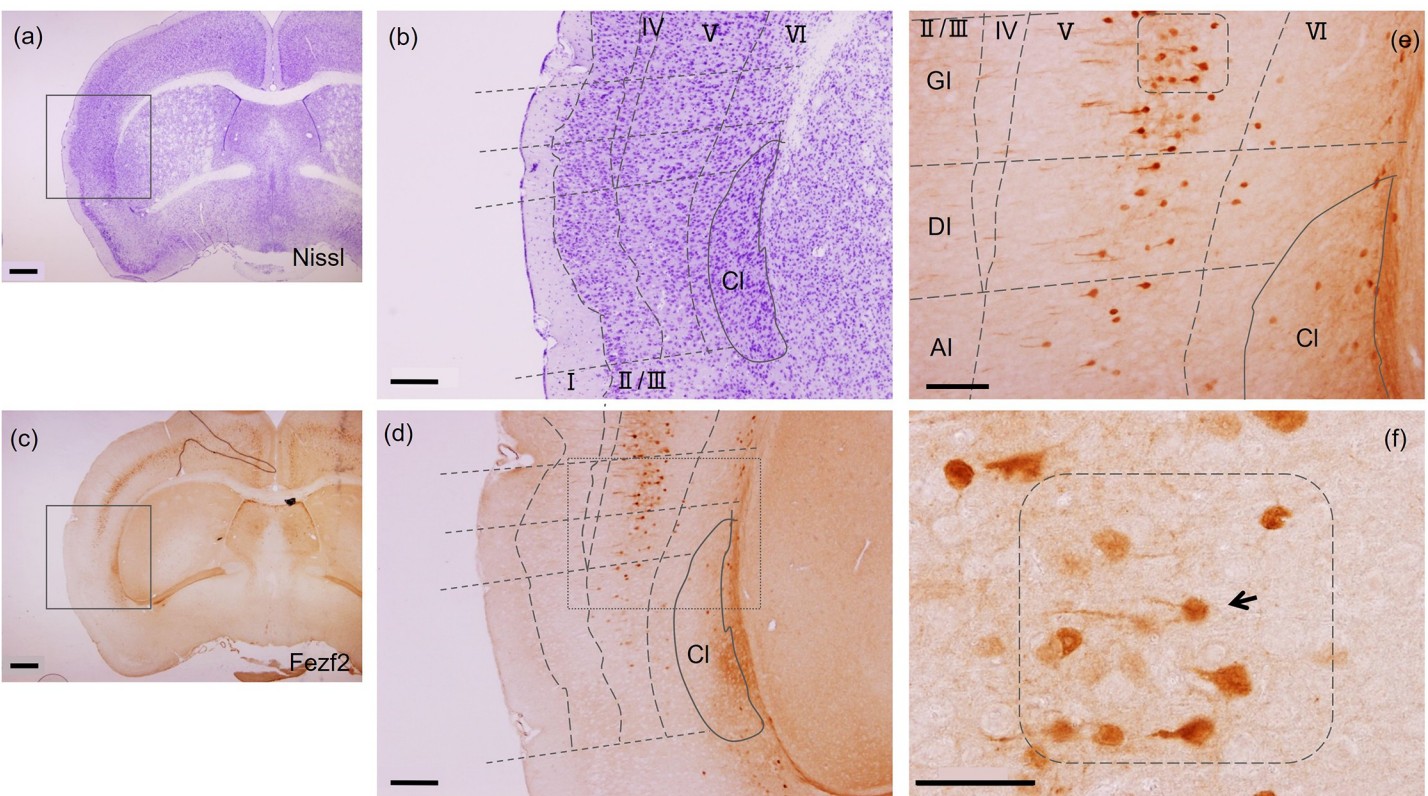

**Fig 2. Fezf2-positive neuron exhibiting fork cell-like shape, having two apical dendrites in layer V neurons of the mouse insular cortex.** A (a), (b), are images taken from the same section Fig 1A4, and (c), (d), (e), (f) are images taken from the same section Fig 1B4. Fezf2 was expressed in layer V and VI with most neurons being densely accumulated in layer V (GI and DI). (a), (b); Nissl staining. (c)—(f); Fezf2. (b), (d) are the higher magnification images of the squares in panels (a), (c), respectively. (e) is the higher magnification image of the square in panel (d). (f) is the enlarged image including the rounded rectangle in panel (e). A Fezf2-positive neuron exhibiting fork cell-like shape, having two apical dendrites, is indicated by an arrow. No VENs-like shaped (bipolar) cells were identified. Scale bars (a), (c): 500 μm, (b), (d): 200 μm, (e): 100 μm, (f): 50 μm.

Fezf2-positive neurons had two apical dendrites that resembled the morphological features of fork cells (Fig 2F). No VEN-like shaped (bipolar) cells were seen in the mouse insular cortex. In order to study further the detailed morphology of Fezf2-positive cells, we performed immunofluorescence for RFP. Signals for RFP were observed in neurons of layers V/VI, with the highest expression in layer V, which was similar with the results shown in Figs 1–3. We again identified a few fork cell-like neurons, which had two apical dendrites, in the mouse insular cortex (Fig 3D and 3E). In human frontoinsular cortex (FI), VENs are often found alongside fork cells characterized by a single large basal dendrite and a bifurcated apical dendrite [11,12]. However, no VEN was observed in the mouse insular cortex analyzed here.

## An immunoelectron microscope established that fork cell-like neurons and/or enveloping cell-like neurons (holding neurons) were found in the mouse insular cortex

Next, to further investigate the fine structure of fork cell-like neurons that were observed in the mouse insular cortex, we employed the immunoperoxidase method for electron microscopy. Intense signals were detected in layers V/VI neurons, with the highest expression in layer V, which was similar with the results shown in Figs 1, 2 and 4. Immunoelectron microscopy established that the fork cell-like neurons that had two apical dendrites and a basal

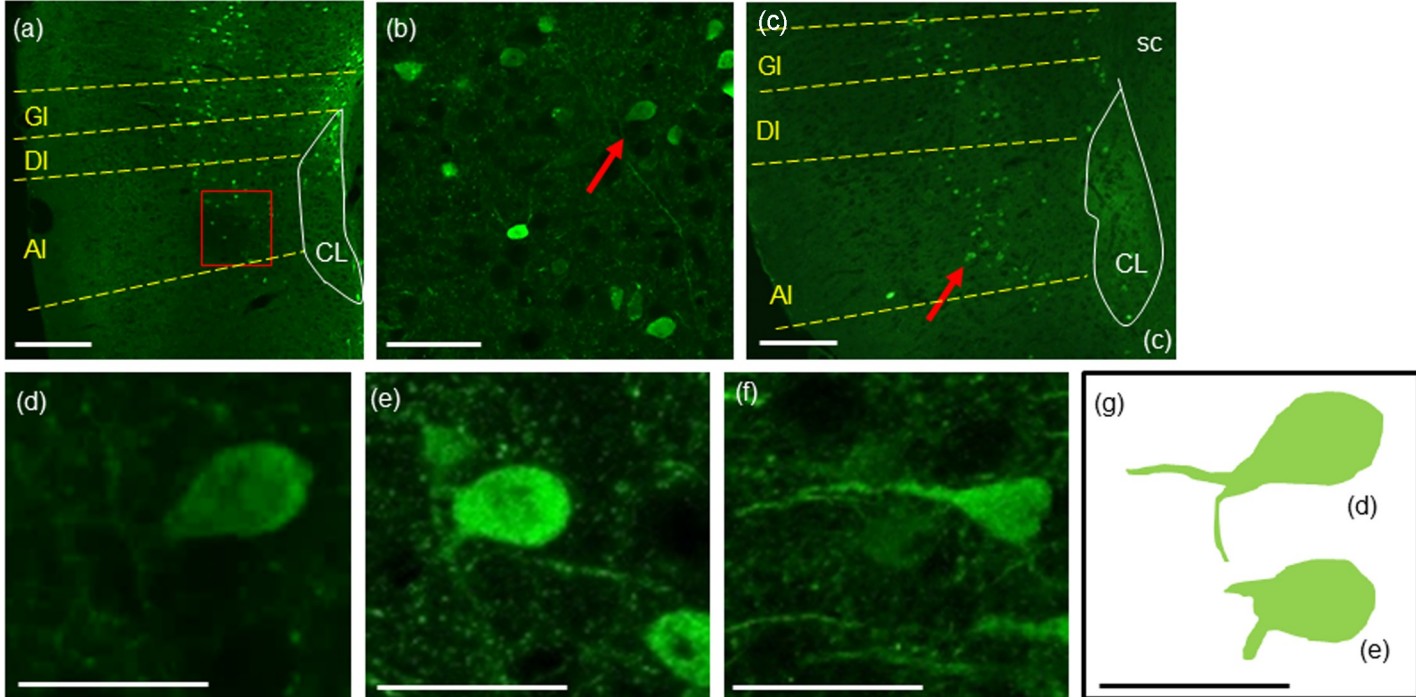

**Fig 3. A few Fezf2-positive neurons in the mouse insular cortex exhibited fork cell-like shape by immunofluorescence staining.** Immunofluorescence against RFP was performed to visualize Fezf2 signals with frozen brain sections of Tg(Fezf2-tdTomato) mouse. (a)—(c) Fezf2 was expressed in layer Ⅴ and Ⅵ. (b) High magnification image of the red squared box in (a). (d) A fork cell-like neuron indicated by the red arrow in (b), which had two apical dendrites. (e) A fork cell-like neuron indicated by the red arrow in (c). (f) A typical pyramidal neuron in (c). No VEN-like shaped (bipolar) cells were found in the mouse insular cortex. (g) A depicted image of neurons in (d) and (e). Scale bars (a), (c): 200 μm, (b): 100 μm, (d)—(g) 20 μm.

dendrite, based on their morphology, were found in the mouse insular cortex (Fig 4Ad). Typical fork cells with bifurcated apical dendrites are thoroughly described by von Angela Syring [33]. Some literature have defined the neuron that embraces a capillary or a neighboring cell as enveloping neuron [27]. In order to discuss precisely about the nature of such fork cell-like neurons, from now on, we here name a neuron that embraces a neighboring capillary an 'enveloping neuron', whereas one that holds a neighboring cell a 'holding neuron'. We identified a few holding neurons in the mouse insular cortex (Fig 4Bc). The nuclear envelopes of the holding neurons had an irregular contour (Fig 4Bc and 4Bd). Electron microscopic analyses showed that the cells embraced by the holding neurons had an electron-lucent nucleus and non-condensed euchromatin (Fig 4Bc).

Next, to consolidate our findings, we reconstructed 3D images of fork cell-like neurons that were observed in the mouse insular cortex. We processed Z-axis image stacks for reconstruction. Neurons that had two apical dendrites were clearly recognized (S1 Movie).

## GRP and NMB were expressed differently in the mouse insular cortex

Next, since GRP and NMB are expressed in VENs and fork cells in human (Table 1), we studied the distribution of *GRP-* or *NMB*-expressing cells in three subregions: GI, DI and AI of the mouse insular cortex. *GRP* or *NMB* was expressed in different parts of the three aforementioned subregions. *GRP* was expressed widely in layer Ⅱ/Ⅲ and layer Ⅴ, with the highest expression in AI (Figs 5 and 6). Alternatively, *NMB* was scattered in the deep layers, layer Ⅴ and layer Ⅵ (GI, DI, and AI), with the highest expression in layer Ⅵ (Figs 5 and 7). Relatively strong *NMB* expression was also observed in the claustrum (Fig 5). Next, we investigated whether *GRP* and

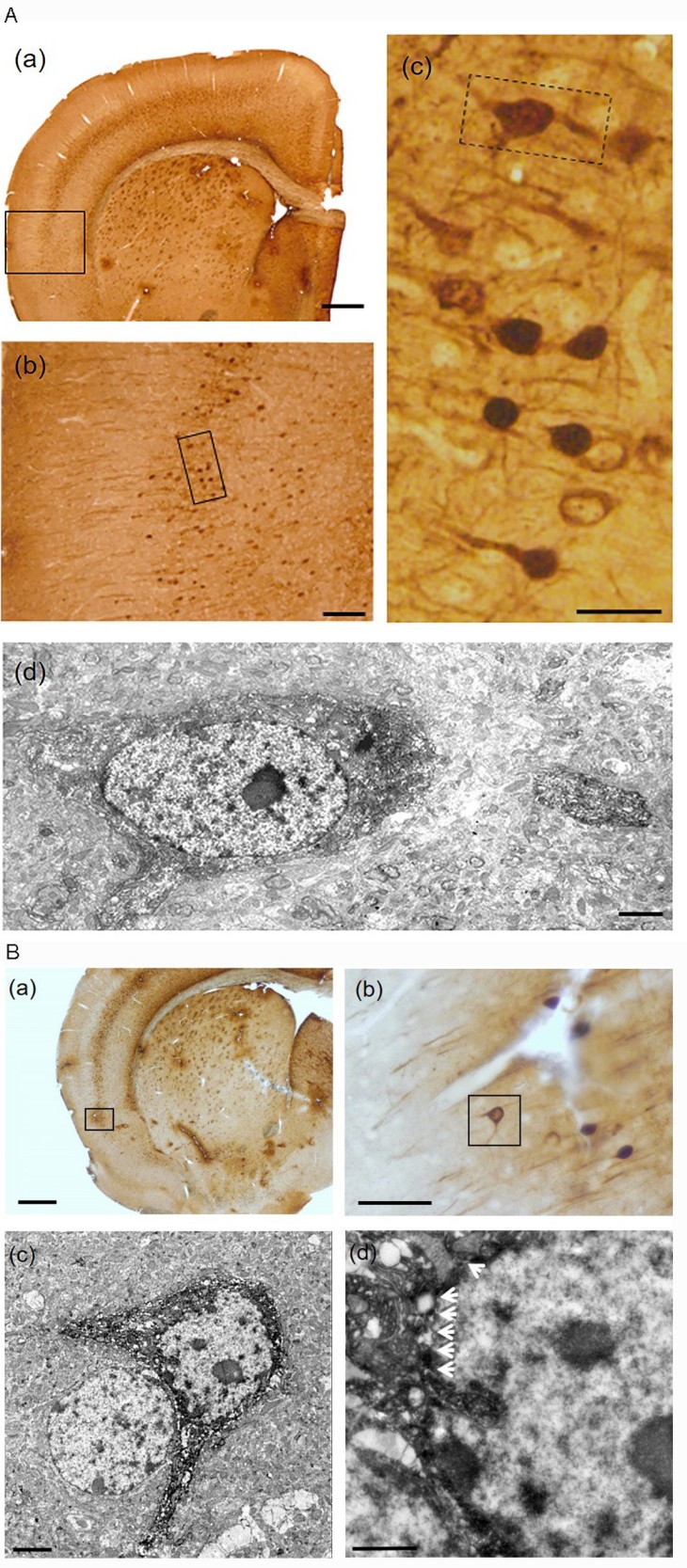

**Fig 4. The ultrastructural analyses of fork cell-like neurons and holding neurons by immunoelectron microscopy.** IHC against RFP was performed to visualize Fezf2 signals with a brain section of Tg(Fezf2-tdTomato) mouse. A. A few Fezf2-positive neurons exhibited fork cell-like neurons that has two apical dendrites and a basal dendrite. (a) Fezf2 was expressed in layer Ⅴ and Ⅵ. (b) High magnification image of the black squared box in (a). (c) High magnification image of the black squared box in (b). A fork cell-like neuron is indicated by the black dotted squared box. (d) Electron microscopy image of the black dotted squared box in (c). The ultrastructure of a fork cell-like neuron, which had two apical dendrites and a basal dendrite were observed in the mouse insular cortex. Scale bars (a): 500 μm, (b): 200 μm, (c): 50 μm, (d) 2 μm. B. A few Fezf2-positive neurons in the mouse insular cortex exhibited enveloping cell-like shape. (a) Fezf2 was expressed in layer Ⅴ and Ⅵ (the black squared box). (b) High magnification image of the black squared box in (a). (c) Electron microscopy image of the black squared box in (b). A holding neuron (see text for further definition), which had two apical dendrites were found in the mouse insular cortex. A cell which was embraced by a holding neuron had weak electron density. (d) High magnification image of (c). The nuclear envelope, having an irregular contour is indicated by arrows. Scale bars (a): 500 μm, (b): 50 μm, (c): 2 μm, (d) 1 μm.

Fezf2 were co-expressed in some neurons of the mouse insular cortex. Once the ISHH signal detection against *GRP* was completed with frozen brain sections of Tg(Fezf2-tdTomato) mouse, we performed immunohistochemistry using antibody against RFP. As is shown in S2 Fig., *GRP* and Fezf2 were co-expressed in some neurons of the mouse insular cortex (S2C Fig).

## Other molecular characteristics

Table 1 is the list of molecules that are reported to be expressed in VENs and/or fork cells in human insular cortex. We additionally examined distribution of these molecules with the Allen Brain Atlas mouse ISHH database. As in humans, *ADRA1A* is expressed in layers II, III and Ⅴ of the mouse insular cortex, although we did not identify any cells morphologically resembling VENs or fork cells. Expression of *VMAT2* or *GABRQ* was not observed in the mouse cerebral cortex.

## Discussion

The major findings of this study are: (1) discovery of fork cell-like neurons in the mouse insular cortex and (2) identification of *NMB*-, *GRP*- or Fezf2-positive neurons in layer V of the mouse insular cortex, which are the molecules that are known to be contained in human VENs and/or fork cells. It has been argued that VENs and/or fork cells do not exist in rodents [17]. On the other hand, Raghanti et al. reported that VENs and fork cells exist in cetaceans, artiodactyls, and perissodactyls with species-specific differences in distributions and cell densities [16]. This suggested that VENs and fork cells were not unique to highly encephalized or socially complex species. In this study, we found several Fezf2-positive neurons exhibiting fork cell-like shape, having two apical dendrites. In addition, although no neurons that morphologically resembled VEN were observed, some possessed *NMB* and/or *GRP*. Therefore, it is possible that equivalents of the fork cells are present in the mouse.

Fezf2 is essential to the differentiation of layer V neurons that project to the striatum, superior colliculus, pons, and spinal cord [34,35]. It is reported that Fezf2 is a marker for the layer V projection neurons in the mouse motor cortex [34]. Earlier evidence of VENs are subcortically projecting neurons [10,19,36] has been further supported by recent findings that VENs preferentially express transcription factors, including Fezf2, which are characteristic of subcortical projection neurons [32]. Our study revealed Fezf2 was localized in certain population of neurons in deep layers of the mouse insula, with the highest expression in layer V. Therefore, it is likely that Fezf2-positive fork cell-like neurons in layer V of the mouse insular cortex are subcortically projecting neurons.

We found that *NMB* and *GRP* were expressed differently in three subregions of the mouse insular cortex; *NMB* was present in a very restricted population of neurons in the deep layers

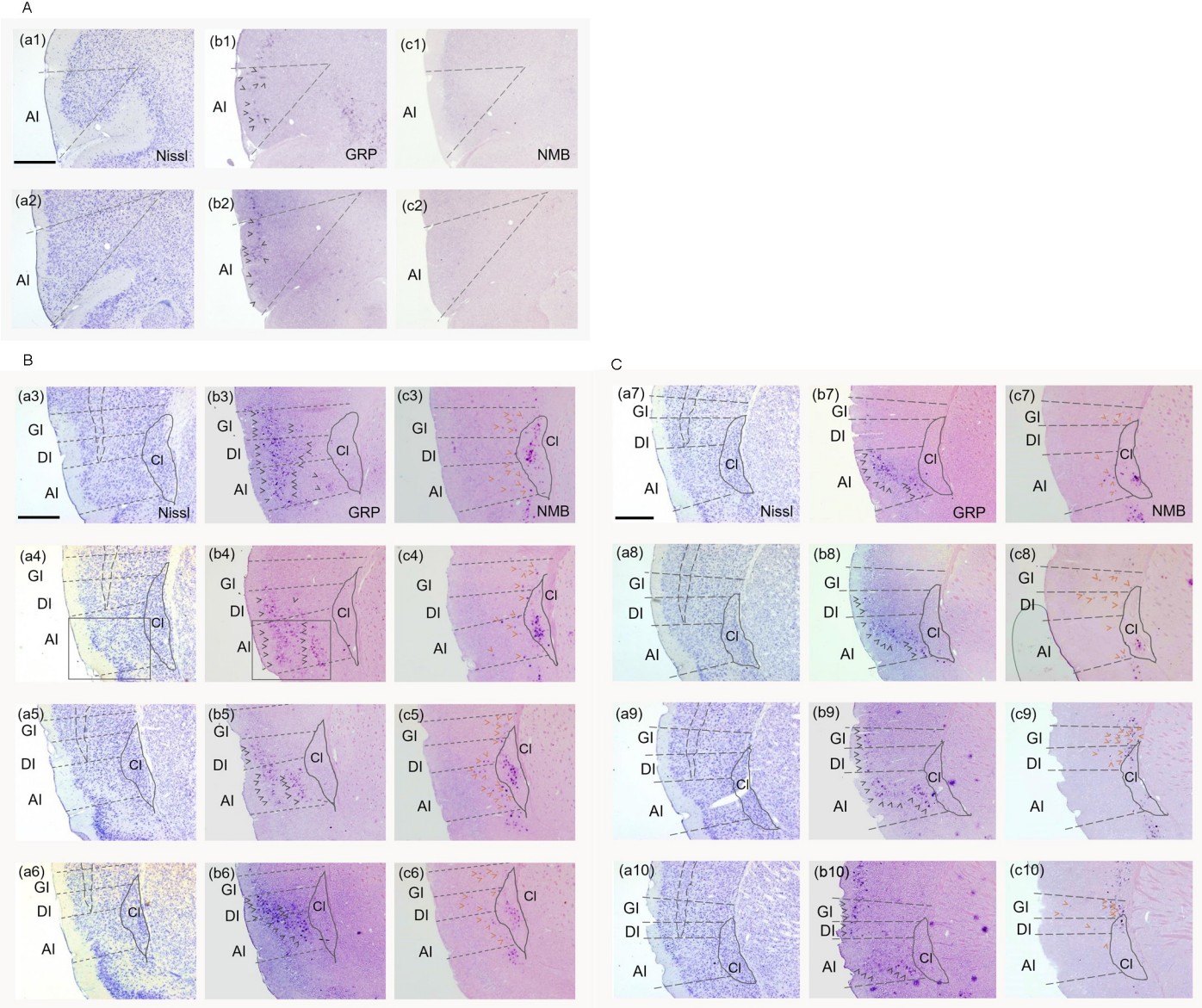

**Fig 5. *GRP* and *NMB* were expressed differently in the mouse insular cortex.** A-C. Distribution of *GRP* or *NMB* in the mouse insular cortex. Sections were numbered in rostral-caudal order. (a1)—(a10) are Nissl-stained adjacent tissue sections of (b1)—(b10) and (c1)—(c10). Gray arrowheads indicate *GRP*-expressing neurons. Red brown arrowheads indicate *NMB*-expressing neurons. Scale bar: 200 μm.

of the insular cortex and the claustrum, but not in the medially adjacent putamen; whereas *GRP* was expressed mainly in layer II/III and layer V of agranular regions in the mouse insular cortex. It is possible that *NMB*-positive neurons and *GRP*-positive neurons belong to different subpopulations in the mouse insular cortex.

Recently Banovac et al. discussed that the identification of specialized cells such as VENs in non-primates should be done by demonstrating the dendritic and axonal morphology or by identifying specific markers or marker combinations that would enable the identification of VENs without relying solely on morphology [37]. We here provided the existent case. We found both *NMB*- and *GRP*-positive neurons in layer V of the mouse insular cortex, suggesting that some neurons in this area have the characteristics of VENs and/or fork cells, despite they

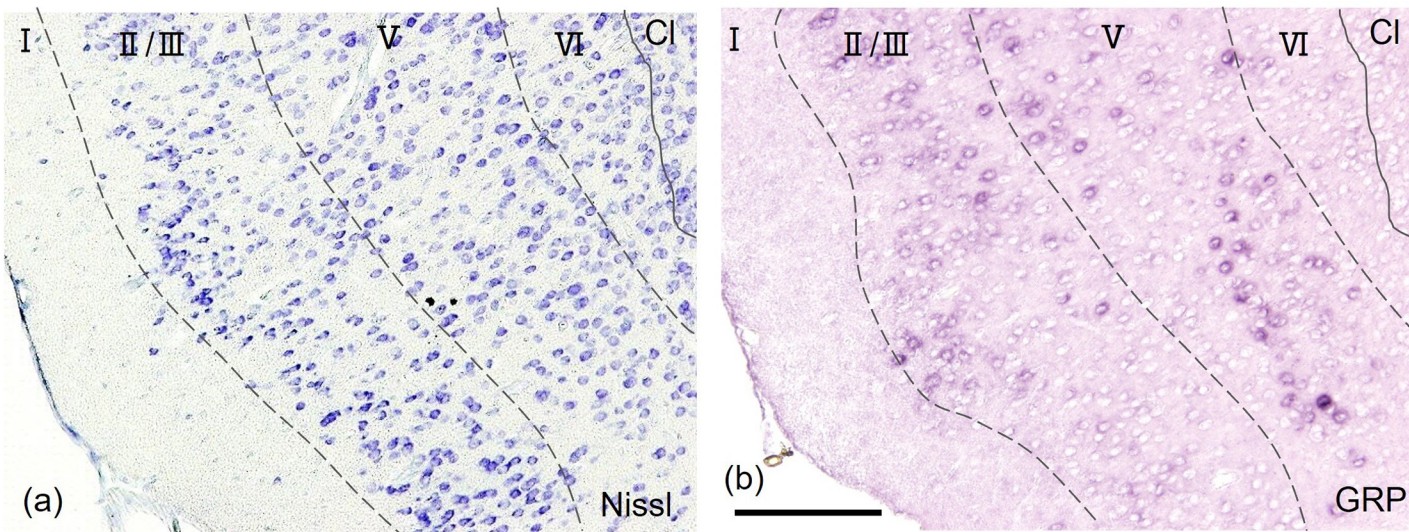

**Fig 6. *GRP* was expressed widely in layer II/III and layer V, with the highest expression in AI of the mouse insular cortex.** (a), (b). High-magnification image of the squared box in Fig 5Ba4 and 5Bb4. (b). *GRP* was expressed widely in layer II/III and layer V. (a) and (b) are adjacent sections. Scale bar: 100 μm.

**Fig 7. *NMB* was expressed in the deep layers, layer V and layer VI (GI, DI, and AI), with the highest expression in layer VI neurons of the mouse insular cortex.** *NMB* expression was scattered in layer V and VI, with the highest expression in layer VI. (d) and (e) are the higher magnification images of the rounded square boxes in (c) and (d). A square in Nissl stained (a) is shown as (b). (b) and (c) are adjacent sections. Scale bars (a): 500 μm, (b), (c): 200 μm, (d) 100 μm, (e): 50 μm.

are not identifiable as VENs and fork cells based on their morphologies. The potential of these mRNAs as VEN and/or fork cell marker should be further investigated.

We defined 'holding neurons' here as those neurons that embrace other cells, but not capillaries. It is possible that the cells embraced by these holding neurons may contribute to new features through interactions by either the cell bodies and/or dendrites of holding neurons, yet further studies are required to clarify the characteristics and biological significance of these holding neurons.

## Supporting information

**S1 Fig. The 3D images.** Some Fezf2-positive neurons in the mouse insular cortex exhibited fork cell-like shape. Immunofluorescence against RFP was performed to visualize Fezf2 signals with brain sections of Tg(Fezf2-tdTomato) mouse. (a) Fezf2 was expressed in layer V and VI. (b) High-magnification image of a fork cell-like neuron indicated by the white arrow in the insular cortex of (a), which had two apical dendrites. Scale bars (a): 100 μm, (b): 10 μm.
(TIF)

**S2 Fig. *GRP* and Fezf2 were co-expressed in the mouse insular cortex.** (a) IHC for RFP was performed after *in situ* hybridization against *GRP*. Tg(Fezf2-tdTomato) mouse. Scale bar: 200 μm. (b) A high-magnification image of (a). Scale bar: 50 μm. (c) A high-magnification image of (b). The black arrow indicates a neuron co-expressing *GRP* and Fezf2. Scale bar: 25 μm.
(TIF)

**S1 Movie. An example of a Fezf2-positive neuron which exhibited fork cell-like shape.** See S1B Fig.
(MP4)

## Acknowledgments

We are grateful to S. Yasumura, K. Kuroda, M.-J. Xie, Y. Mori, and T. Iguchi for helpful discussion, A. Emi for technical assistance, and M. Yamaguchi, Y. Shibuya, K. Danke and A. Yoshinori for secretarial assistance. The experiments were supported by Center for Medical Research and Education, Graduate School of Medicine, Osaka University. Animal experiments were supported by The Institute of Experimental Animal Science, Faculty of Medicine, Osaka University.

## Author Contributions

**Conceptualization:** Makoto Sato.

**Data curation:** Manabu Taniguchi.

**Formal analysis:** Manabu Taniguchi.

**Funding acquisition:** Makoto Sato.

**Investigation:** Manabu Taniguchi, Misaki Iwahashi, Makoto Sato.

**Methodology:** Manabu Taniguchi, Yuichiro Oka, Sheena Y. X. Tiong, Makoto Sato.

**Project administration:** Makoto Sato.

**Resources:** Yuichiro Oka, Makoto Sato.

**Supervision:** Makoto Sato.

**Validation:** Manabu Taniguchi, Makoto Sato.

**Visualization:** Manabu Taniguchi, Yuichiro Oka, Sheena Y. X. Tiong, Makoto Sato.

**Writing – original draft:** Manabu Taniguchi, Makoto Sato.

**Writing – review & editing:** Yuichiro Oka, Sheena Y. X. Tiong, Makoto Sato.

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
