## [Decision Letter · Decision Letter 0]

11 Apr 2022

PONE-D-22-05313Fezf2-positive Fork cell-like neurons in the mouse insular cortexPLOS ONE

Dear Dr. Sato,

Thank you for submitting your manuscript to PLOS ONE. After careful consideration, we feel that it has merit but does not fully meet PLOS ONE’s publication criteria as it currently stands. Therefore, we invite you to submit a revised version of the manuscript that addresses the points raised during the review process.

ACADEMIC EDITOR: This academic editor think that, especially, more detailed morphological analysis is required as pointed out by Reviewer 2.

We look forward to receiving your revised manuscript.

Kind regards,

Masabumi Minami, Ph.D.

Academic Editor

PLOS ONE

Journal Requirements:

Reviewers' comments:

Reviewer's Responses to Questions

**Comments to the Author**

1. Is the manuscript technically sound, and do the data support the conclusions?

Reviewer #1: Yes

Reviewer #2: Partly

2. Has the statistical analysis been performed appropriately and rigorously? 

Reviewer #1: Yes

Reviewer #2: N/A

3. Have the authors made all data underlying the findings in their manuscript fully available?

Reviewer #1: Yes

Reviewer #2: Yes

4. Is the manuscript presented in an intelligible fashion and written in standard English?

Reviewer #1: Yes

Reviewer #2: Yes

5. Review Comments to the Author

Reviewer #1: The study by Taniguchi et al. （MS# PONE-D-22-05313）tested the presence in the mouse insular cortex of an equivalent of the von Economo neuron and/or folk cell, which are reported in the insular cortex and anterior cingulate cortex of higher animals and possess characteristic dendritic morphology and molecular expression, such as Fezf2, neuromedin B, and gastrin-releasing peptide. To do this, authors examined the morphology of Fezf2-expressing neurons by the light and electron microscopic levels, and identified such neurons that have two characteristic apical dendrites and embrace neighboring different neurons. Moreover mRNAs for neuromedin B, and gastrin-releasing peptide were expressed in this brain region. Based on these findings, authors conclude that it is likely that neurons equivalent to folk cells are present in the mouse insular cortex.

This paper is well organized and written, and experiments are appropriately executed. I raise one major comment only.

Major

1. At the regional level, authors showed the presence of neurons expressing Fezf2, neuromedin B mRNA, and gastrin-releasing peptide mRNA in the mouse insular cortex. However, it is still unclear whether or to which extents Fezf2-positve neurons express these peptides. This can be answered by combining ISH for these peptides with immunohistochemistry for Fezf2 (RFP), and should strengthen the conclusion of this study.

Minor

2. In method sections for Tissue Preparation, anesthetize mice were intracardially perfused by either 4% PFA or 0.05%GLA/4% PFA. If fresh frozen sections were used for ISH experiment, additional information on the preparation of fresh frozen section from anesthetized mice is necessary. Please confirm this point.

3. Reported cell maker molecules for human von Economo neurons and/or folk cells are listed in Table 1. Based on this, authors selected Fezf2 to examine expression profiles in the mouse insular cortex. It is better to be explained why Fezf2 is selected from the listed molecules.

Reviewer #2: The authors addressed von Ecomono neurons (VEN) and the folk cells (FC) whose existence in cerebral cortex are known specifically in human, great ape and some evolutionally higher animals. In this paper the authors demonstrated possible existence of the FC in rodent cerebral cortex as well, and attempted to compare molecular characteristics between human and mice.

The major concern would be the morphologies they demonstrated are not fully convincing. Neurons in Figures 2f, 3e and 4c appears to have two apical processes, but how the authors exclude the possible existence of additional major process. Currently researchers are able to capture 3D morphology of cells by using the software such as Imaris or something comparable. Alternatively a sequential scanning EM, which provide 3D images of EM, would be available to show more precise primary dendritic arbor.

Another concern is the significance of Fezf2 in mice FC. Previous paper by Tantirigama et al (2015) demonstrated very clear localization of Fezf2 using Fezf2-Gfp reporter mouse. They identified two distinct subtypes of Fezf2+ neurons in the matured cortex that resembled pyramidal tract projection neurons (PT-PNs) and intratelencephalic projection neurons (IT-PNs). This paper also demonstrated that Fezf2 positive neurons were widely localized in the layer 5 of various cortical regions from the motor to sensory. This would suggest that Fezf2 does not necessarily imply molecular or functional significances in FC.

Another aim of this study, I guess, might be molecular characterization of the FC and VEN enquiring whether the molecular composition in FC is similar or not between human and mice. Human VENs and FC may contain NMB, GRP, Fezf2. However, all these molecules could show much wider localized in the brain. They concluded that the additional possible markers such as NMB, GRP, VMAT2, and GABRQ seem negative in FC, although they need more precise labeling indicating both the two apical processes and immunostaining simultaneously. If the authors found a specific combination of markers are useful for the identification of FC, it would be a good job.

Other minor point:

The authors used ICR strain of mice. Why do authors used this strain rather than C57BL6. This should be mentioned.

As for the holding neurons, how many cells are examined with the EM. Do all the FC cells have this type of cells between the two proximal dendrites.?

6. PLOS authors have the option to publish the peer review history of their article (what does this mean?). If published, this will include your full peer review and any attached files.

Reviewer #1: No

Reviewer #2: No

---

## [Author Response · Author response to Decision Letter 0]

18 Aug 2022

Reviewer #1: We would like to thank Reviewer #1 for valuable comments on our manuscript. I have incorporated all of your suggestions into my revision. They were very helpful. Thank you for your help.

Reviewer #2: We would like to thank Reviewer #2 for valuable comments on our manuscript. I have incorporated all of your suggestions into my revision. They were very helpful. Thank you very much.

---

## [Decision Letter · Decision Letter 1]

24 Aug 2022

Fezf2-positive Fork cell-like neurons in the mouse insular cortex

PONE-D-22-05313R1

Dear Dr. Sato,

We’re pleased to inform you that your manuscript has been judged scientifically suitable for publication and will be formally accepted for publication once it meets all outstanding technical requirements.

Kind regards,

Masabumi Minami, Ph.D.

Academic Editor

PLOS ONE

Additional Editor Comments (optional):

Reviewers' comments:

Reviewer's Responses to Questions

**Comments to the Author**

1. If the authors have adequately addressed your comments raised in a previous round of review and you feel that this manuscript is now acceptable for publication, you may indicate that here to bypass the “Comments to the Author” section, enter your conflict of interest statement in the “Confidential to Editor” section, and submit your "Accept" recommendation.

Reviewer #1: All comments have been addressed

Reviewer #2: All comments have been addressed

2. Is the manuscript technically sound, and do the data support the conclusions?

Reviewer #1: Yes

Reviewer #2: Yes

3. Has the statistical analysis been performed appropriately and rigorously? 

Reviewer #1: Yes

Reviewer #2: Yes

4. Have the authors made all data underlying the findings in their manuscript fully available?

Reviewer #1: Yes

Reviewer #2: Yes

5. Is the manuscript presented in an intelligible fashion and written in standard English?

Reviewer #1: Yes

Reviewer #2: Yes

6. Review Comments to the Author

Reviewer #1: Now authors responded to my previous comments made to the original MS, with no more additional comments.

Reviewer #2: (No Response)

7. PLOS authors have the option to publish the peer review history of their article (what does this mean?). If published, this will include your full peer review and any attached files.

Reviewer #1: No

Reviewer #2: No

---

## [Editor Report · Acceptance letter]

26 Aug 2022

PONE-D-22-05313R1 

Fezf2-positive Fork cell-like neurons in the mouse insular cortex 

Dear Dr. Sato:

I'm pleased to inform you that your manuscript has been deemed suitable for publication in PLOS ONE. Congratulations! Your manuscript is now with our production department. 

Kind regards, 

on behalf of

Dr. Masabumi Minami 

Academic Editor

PLOS ONE